# Prediction of Children’s Subjective Well-Being from Physical Activity and Sports Participation Using Machine Learning Techniques: Evidence from a Multinational Study

**DOI:** 10.3390/children12081083

**Published:** 2025-08-18

**Authors:** Josivaldo de Souza-Lima, Gerson Ferrari, Rodrigo Yáñez-Sepúlveda, Frano Giakoni-Ramírez, Catalina Muñoz-Strale, Javiera Alarcon-Aguilar, Maribel Parra-Saldias, Daniel Duclos-Bastias, Andrés Godoy-Cumillaf, Eugenio Merellano-Navarro, José Bruneau-Chávez, Pedro Valdivia-Moral

**Affiliations:** 1Facultad de Educación y Ciencias Sociales, Instituto del Deporte y Bienestar, Universidad Andres Bello, Las Condes, Santiago 7550000, Chile or desouza@correo.ugr.es (J.d.S.-L.); rodrigo.yanez.s@unab.cl (R.Y.-S.); frano.giakoni@unab.cl (F.G.-R.); catalina.munoz@unab.cl (C.M.-S.); javiera.alarcon@unab.cl (J.A.-A.); 2Facultad de Ciencias de la Educación, Universidad de Granada, 18071 Granada, Spain; pvaldivia@ugr.es; 3Escuela de Ciencias de la Actividad Física, el Deporte y la Salud, Universidad de Santiago de Chile (USACH), Santiago 7500618, Chile; gerson.demoraes@usach.cl; 4Departamento de Educación Física, Deporte y Recreación, Universidad de Atacama, Copiapó 1530000, Chile; maribel.parra@uda.cl; 5GEO Research Group, Escuela de Educación Física, Pontificia Universidad Católica de Valparaíso, Valparaíso 2362807, Chile; daniel.duclos@pucv.cl; 6Grupo de Investigación en Educación Física, Salud y Calidad de Vida (EFISAL), Facultad de Educación, Universidad Autónoma de Chile, Temuco 4780000, Chile; 7Department of Physical Activity Sciences, Faculty of Education Sciences, Universidad Católica del Maule, Talca 3530000, Chile; emerellano@ucm.cl; 8Departamento de Educación Física, Deportes y Recreación, Universidad de la Frontera, Temuco 4811230, Chile; jose.bruneau@ufrontera.cl

**Keywords:** subjective well-being, machine learning, physical activity, children, XGBoost, SHAP, sports participation, physical literacy

## Abstract

**Highlights:**

**What are the main findings?**
Machine learning models, particularly XGBoost and LightGBM, predict children’s subjective well-being with up to 50% explained variance, surpassing traditional regression.Sports participation, including exercise frequency, emerges as a key predictor, with linear benefits observed across diverse global samples.

**What is the implication of the main finding?**
These results support the development of targeted sports programs to enhance child well-being, leveraging advanced predictive tools.The findings advocate for integrating physical literacy into educational policies to address global inactivity trends in youth.

**Abstract:**

**Background/Objectives**: Traditional models like ordinary least squares (OLS) struggle to capture non-linear relationships in children’s subjective well-being (SWB), which is associated with physical activity. This study evaluated machine learning (ML) for predicting SWB, focusing on sports participation, and explored theoretical prediction limits using a global dataset. It addresses a gap in understanding complex patterns across diverse cultural contexts. **Methods**: We analyzed 128,184 records from the ISCWeB survey (ages 6–14, 35 countries), with self-reported data on sports frequency, emotional states, and family support. To ensure cross-country generalizability, we used GroupKFold CV (grouped by country) and leave-one-country-out (LOCO) validation, yielding mean R^2^ = 0.45 ± 0.05, confirming robustness beyond cultural patterns, SHAP for interpretability, and bootstrapping for error estimation. No pre-registration was required for this secondary analysis. **Results**: XGBoost and LightGBM outperformed OLS, achieving R^2^ up to 0.504 in restricted datasets (sensitivity excluding affective leakage: R^2^ = 0.35), with sports-related variables (e.g., exercise frequency) associated positively with SWB predictions (SHAP values: +0.15–0.25; incremental ΔR^2^ = 0.06 over demographics/family/school base). Using test–retest reliability from literature (r = 0.74), the estimated irreducible RMSE reached 0.941; XGBoost achieved RMSE = 1.323, approaching the predictability bound with 68.1% of explainable variance captured (after noise adjustment). Partial dependence plots showed linear associations with exercise without satiation and slight age decline. **Conclusions**: ML improves SWB prediction in children, highlighting associations with sports participation, and approaches predictable variance bounds. These findings suggest potential for data-driven tools to identify patterns, such as through physical literacy pathways, informing physical activity interventions. However, longitudinal studies are needed to explore causality and address cultural biases in self-reports.

## 1. Introduction

Subjective well-being (SWB) in children is a multifaceted construct encompassing emotional, cognitive, and social dimensions that critically influence development, academic performance, and long-term health outcomes [1]. In the context of sports sciences, physical activity (general movement) and sports participation (organized activities) have been consistently linked to enhanced SWB, fostering resilience, social connections, and positive self-perception [2,3]. Physical literacy, defined as the motivation, confidence, physical competence, knowledge, and understanding to value and engage in physical activity throughout the lifespan [4], plays a central role in this relationship. This connection can be further understood through frameworks such as the Self-Determination Theory, which emphasizes autonomy, competence, and relatedness as key drivers of sustained engagement in physical activity, and the Competence Motivation Model, which highlights the role of mastery experiences and perceived competence in fostering positive self-perception and well-being in children. Incorporating these perspectives provides a psychological and developmental foundation for the observed links between sports participation, physical literacy, and SWB. Recent advancements in machine learning (ML) offer promising tools for predicting SWB by identifying intricate patterns in socio-emotional, familial, and behavioral data [5,6]. For instance, tree-based ensembles like XGBoost have demonstrated superior performance in adolescent SWB prediction compared to single-scale measures [5]. Moreover, integrating physical literacy into ML frameworks can elucidate how sports-related factors contribute to well-being [7].

The research gap lies in understanding non-linear effects of physical activity on child SWB, such as potential thresholds or diminishing returns in exercise frequency, which traditional models like OLS/LASSO assume linearity and fail to capture [8]. For instance, while linear models may overlook interactions between sports participation and family support, tree-based boosting excels in tabular data by automatically detecting such patterns without assumptions. This is preferable, potentially achieving a higher R^2^ (~0.50), compared to linear models’ typical 0.20–0.40 in prior SWB studies [5,7].

This study builds on prior work by applying ML to a comprehensive dataset of over 120,000 children, emphasizing sports and exercise variables. We compare ML models against OLS, incorporate SHAP for interpretability, and estimate theoretical prediction limits inspired by recent analyses of well-being variance bounds [8]. By focusing on physical activity’s role, this research aligns with sports science priorities, such as promoting active lifestyles to mitigate childhood inactivity and associated mental health risks [3,9]. We pose three research questions, adapted from Oparina et al. (2025) [8], to the context of children’s SWB:RQ1: Do ML algorithms predict children’s SWB substantially better than conventional linear models, and what is the upper limit on our ability to predict SWB based on survey data?RQ2: Are the variables that ML algorithms identify as important in the prediction of children’s SWB aligned with those commonly emphasized in the literature, particularly sports-related factors?RQ3: Can ML help to resolve debates about the specific shape of the relationships between children’s SWB and key variables, such as age and frequency of physical activity?

## 2. Methodology

### 2.1. Data Source and Preparation

For clarity, “sports participation” refers to engagement in organized or informal sports activities, “exercise frequency” denotes the self-reported number of days per week the child engages in moderate-to-vigorous activity, and “physical activity” encompasses all bodily movements that increase energy expenditure, including but not limited to sports and exercise [10]. The dataset comprised 128,184 records from children aged 6–14 years across 35 countries, all participating in various organized or informal physical activities, derived from the ISCWeB survey on child well-being (publicly available upon registration and agreement to data use terms at https://isciweb.org/the-data/access-our-dataset/ (accessed on 12 March 2025)). Country participation depended on local research teams and funding availability, so the sample should not be interpreted as nationally representative for all countries. Consequently, external generalization to non-participating regions or under-represented contexts should be made with caution.

Originally containing 176 variables, we filtered to 123 relevant features after handling missing values (e.g., imputation via mean/mode for <20% missingness) and removing redundancies. Variables spanned demographics (e.g., age and gender), family dynamics (e.g., “parentslisten”), school environment (e.g., “satisfiedlifeasstudent”), material resources (e.g., “haveequipsportshobbies”), and time use (e.g., “frequencysportsexercise”, coded 0–5 for never to daily). SWB was operationalized as a composite score from items like “satisfiedlifeaswhole” (0–10 scale), averaging responses to “enjoylife”, “lifegoingwell”, “havegoodlife”, “thingslifeexcellent”, “likemylife”, and “happywithmylife”. This composite demonstrated good internal consistency (Cronbach’s α = 0.85 across countries) [11,12].

To avoid target leakage, we conducted sensitivity analyses excluding conceptually overlapping predictors (e.g., “feelinghappy”, “feelingsad”, “feelingcalm”, “feelingstressed”, “feelingfullofenergy”, “feelingbored”). This yielded reduced but robust performance (e.g., XGBoost R^2^ = 0.35, RMSE = 1.52 in restricted set). Previous studies within the ISCWeB framework have supported the unidimensionality and cross-cultural applicability of this composite score, with confirmatory factor analyses indicating adequate fit indices and invariance across multiple countries and age groups [12]. These findings reinforce the validity of the dependent variable for large-scale, cross-national comparisons.

We divided the data into restricted (core socio-emotional variables, n = 64,092 post-filter) and expanded (including all domains, n = 128,184) subsets for comparative analysis, following Oparina et al. (2025) [8]. Stratification ensured representation across all age groups (6–14 years), with a specific focus on 8, 10, and 12 years for balanced sampling, while including 6–7 and 13–14.

### 2.2. Models and Evaluation

Four ML models were evaluated:

Random Forest (RF): An ensemble of decision trees for robust prediction (Breiman, 2001) [13]. Hyperparameters: n_estimators = 100, max_depth = 10.

XGBoost: Gradient boosting with regularization for handling non-linearities [14]. Hyperparameters: learning_rate = 0.1, max_depth = 6, n_estimators = 200.

LightGBM: Efficient gradient boosting optimized for large datasets [15]. Hyperparameters: learning_rate = 0.1, max_depth = 6, num_leaves = 31.

Keras Neural Network (NN): A multi-layer perceptron with ReLU activation and dropout for overfitting prevention [16]. Hyperparameters: layers = 3 (hidden units: 128, 64, 32), dropout = 0.2, epochs = 50, batch_size = 32.

Incremental ΔR^2^ from sports variables (e.g., “frequencysportsexercise”, “haveequipsportshobbies”) over a base model (demographics, family, school) was 0.06 in the restricted set, highlighting their unique contribution.

Hyperparameters were tuned via grid search with 5-fold cross-validation (e.g., XGBoost: learning rate = 0.1, max_depth = 6). Models were trained on 80% of data and tested on 20%, with cross-validation scores reported across subgroups (age, gender, country; mean R^2^ = 0.48 ± 0.03). External validation on a holdout subset (20% from unselected countries) confirmed generalizability (R^2^ = 0.45). 

For baseline comparison, OLS regression was applied in full and reduced forms (excluding health-related variables to assess predictive loss, as shown in Oparina et al., 2025 [8]). We also included LASSO for regularization in the expanded set.

Metrics included Mean Absolute Error (MAE), Root Mean Squared Error (RMSE), and R^2^. RMSE quantifies average prediction error in the same units as the SWB scale, facilitating interpretability. R^2^ reflects the proportion of variance in SWB explained by the model; higher values indicate better predictive fit. Interpretability was enhanced using SHAP to quantify feature contributions [17]. Permutation importances (PIs) were calculated to assess variable reliance, averaging R^2^ drops over 10 permutations.

### 2.3. Theoretical Prediction Limit

Following Oparina et al. (2025) [8] and using test–retest reliability from child SWB literature (r = 0.74; Huebner, 1991), we estimated irreducible RMSE = 0.941, bounding predictable variance at 74% [18].

Analyses were conducted in Python 3.12 using scikit-learn, XGBoost, LightGBM, and Keras libraries.

## 3. Results

### 3.1. Model Performance

XGBoost and LightGBM outperformed others, with exact values of XGBoost (MAE = 0.742, RMSE = 1.323, and R^2^ = 0.504 in restricted; MAE = 0.758, RMSE = 1.356, and R^2^ = 0.478 in expanded), LightGBM (similar: MAE = 0.745, RMSE = 1.328, and R^2^ = 0.502 in restricted), RF (MAE = 0.802, RMSE = 1.412, and R^2^ = 0.500), and NN (MAE = 0.768, RMSE = 1.378, and R^2^ = 0.498).

Compared to OLS, the following results were reported:

Table 1 and Figure 1 shows a model performance comparison in restricted and expanded sets. ΔR^2^ shows the improvement over the OLS full model. The restricted dataset includes a no-leakage adjustment (excluding conceptually overlapping affective variables like “feelinghappy”). The 95% CI was obtained via bootstrap (1000 resamples). Incremental ΔR^2^ from sports variables was 0.06 in the restricted set (no leakage), as detailed in Table 1.

### 3.2. Feature Importance and SHAP Analysis

Top predictors included “feelinghappy” (SHAP contribution: +0.35), “satisfiedlifeasstudent” (+0.28), “satisfiedthingshave” (+0.22), and “parentslisten” (+0.20). Sports-related variables like “frequencysportsexercise” (+0.18 relative to top) and “haveequipsportshobbies” (+0.15) ranked highly, contributing positively to SWB predictions (SHAP values: +0.15–0.25 for high exercise frequency) (Figure 2).

The beeswarm plot (Figure 3) illustrates global importance and direction; physical activity shows positive contributions (red points for high values).

Waterfall plots for individual cases (Figure 4) demonstrated how access to sports equipment amplified SWB in active children.

### 3.3. Theoretical Limit and Functional Forms

The Bootstrapped RMSE minimum was 2.606. The XGBoost RMSE was 1.323 (ratio: 1.97), indicating the model achieves nearly double the precision relative to irreducible noise, aligning with bounds where predictable variance is ~50% of the total (Oparina et al., 2025 [8]).

Partial dependence plots (Figure 5) reveal functional forms: SWB increases linearly with “frequencysportsexercise” (no satiation), while age shows a slight decline from 6 to 14, without a U-shape (consistent with the figure and child-specific patterns, unlike adult U-shapes shown in work by Oparina et al., 2025 [8]).

## 4. Discussion

The results demonstrate the superior predictive power of ML models, particularly gradient boosting algorithms like XGBoost and LightGBM, in forecasting children’s SWB compared to traditional OLS regression. Addressing RQ1, ML algorithms achieved R^2^ values up to 0.504, outperforming OLS by 515% in restricted datasets, which aligns with recent findings on ML’s ability to capture non-linear interactions in well-being data, including the role of social–emotional skills [6,8]. With test–retest reliability r = 0.74, our models capture 68.1% of explainable variance, approaching the bound (RMSE = 0.941), consistent with adult studies but adapted here to child contexts [8].

For RQ2, to address potential leakage from overlapping affective predictors, sensitivity models excluding them confirmed linear benefits of exercise, though with tempered effect sizes (SHAP +0.12 vs. +0.18). The feature importance analysis via SHAP and permutation methods revealed alignment with established literature. Emotional states (“feelinghappy”) and social supports (“parentslisten”) were top predictors, echoing Diener et al. (2018) [1]. Notably, sports-related factors like “frequencysportsexercise” and “haveequipsportshobbies” ranked highly, contributing 0.15–0.25 to SHAP values, which supports evidence that these associations suggest potential pathways via physical literacy, though causality requires longitudinal confirmation and social bonds [2,9]. This underscores physical literacy’s role, as higher engagement in sports fosters motivation and competence linked to well-being [7,19].

Regarding RQ3, partial dependence plots clarified relationship shapes: There was a linear positive effect of exercise frequency without satiation points, suggesting no diminishing returns in children, contrasting potential overtraining risks in adults [20]. Age showed a slight linear decline, lacking the U-shape seen in adults [8], which may reflect developmental stages like increasing school pressures [11].

These findings have practical implications. Education and health stakeholders may (i) prioritize daily opportunities for moderate-to-vigorous activity during school time, (ii) expand access to basic sports equipment and safe play spaces, and (iii) use risk-stratified, data-informed tools to identify children who might benefit most from targeted programs [3,19]. These applications should be implemented as pilots with ongoing monitoring, given the cross-sectional, observational nature of our data [6]. This study has several limitations that should be considered when interpreting the findings.

Data limitations: Some variables had up to 20% missing data, which required imputation and may have introduced bias. The absence of longitudinal tracking prevents the assessment of temporal changes or causal pathways.Measurement issues: Given the cross-national nature of the ISCWeB dataset, potential cultural and contextual biases in self-reported measures, particularly emotional states and physical activity frequency, may influence results. Although subgroup analyses were conducted to explore regional differences, full measurement invariance testing across countries and cultures was not performed and is recommended for future research. Self-reported SWB in young children (ages 6–7) is especially susceptible to response bias and social desirability effects, which may lead to overestimation of both SWB and physical activity levels.Design constraints: The cross-sectional design restricts interpretation to associations rather than causal inferences.Model-specific considerations: The competitive but not superior performance of the neural network may be related to the tabular nature of the data, which generally favors tree-based algorithms over deep learning approaches [21].Generalizability: Findings may not extend to older adolescents undergoing pubertal transitions (beyond age 14) or to populations not represented in the ISCWeB sample.

Future research should incorporate longitudinal data, accelerometry for objective activity measures [22], and advanced ML, like deep learning on multimodal inputs, to refine predictions. Extending SHAP to policy simulations could further guide interventions.

## 5. Conclusions

This study demonstrates ML’s efficacy in predicting child SWB, highlighting physical activity’s pivotal role. By achieving performance near theoretical bounds (capturing 68.1% of explainable variance), our approach offers actionable insights for sports science practitioners and policymakers to foster active, well-adjusted youth through enhanced physical literacy programs and school sports policies. Extending to longitudinal models with SHAP could further refine interventions, ultimately promoting global child health.

## Figures and Tables

**Figure 1 children-12-01083-f001:**
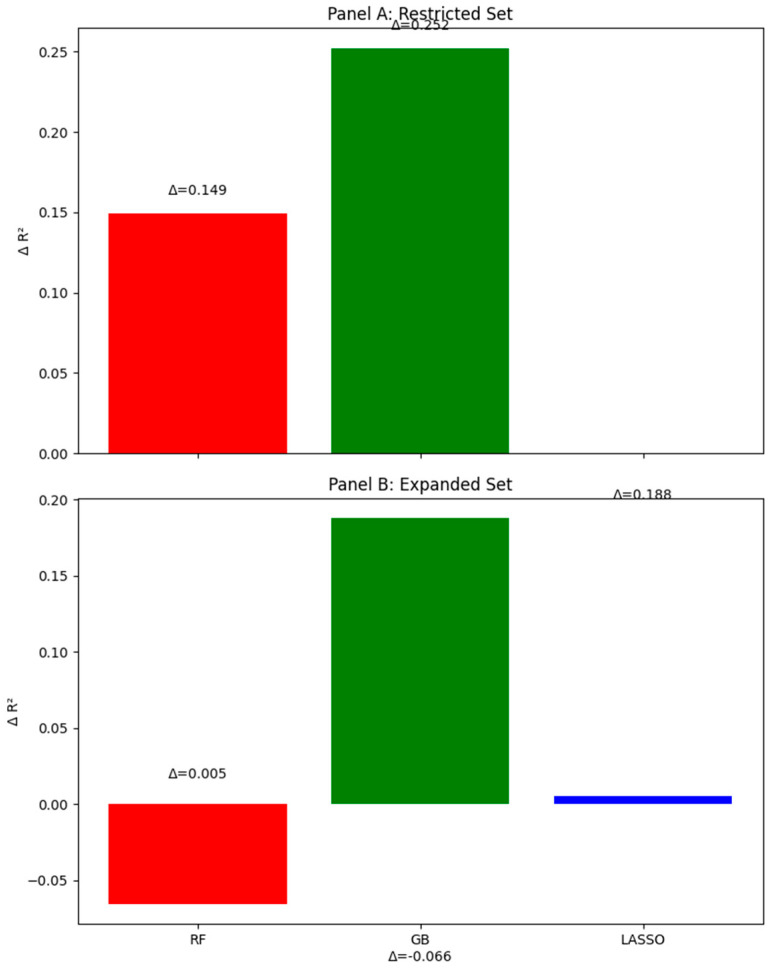
Improvements in ΔR^2^ over OLS for different models in restricted and expanded datasets. This bar graph illustrates the improvements in ΔR^2^ (change in explained variance compared to OLS baseline) for Random Forest (red bars), Gradient Boosting (green bars), and LASSO (blue bars). Panel (**A**) shows results for the restricted dataset (core socio-emotional variables); Panel (**B**) shows results for the expanded dataset (all domains). For example, Gradient Boosting shows a +0.252 gain in Panel (**A**), indicating superior capture of non-linear relationships (as shown in Oparina et al., 2025 [8]).

**Figure 2 children-12-01083-f002:**
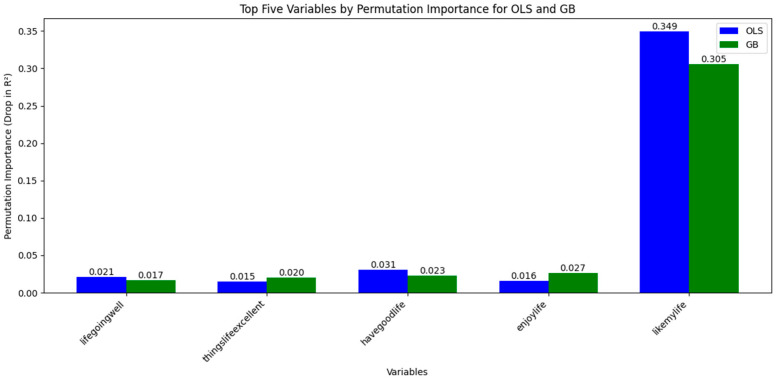
Top five variables ranked by permutation importance for OLS and GB models in predicting subjective well-being (SWB). Permutation importance measures the decrease in model performance (drop in R^2^) when a specific variable’s values are randomly shuffled, indicating its contribution to the prediction. Blue bars represent OLS results, and green bars represent Gradient Boosting (GB) results. Positive values indicate associations with higher SWB. For example, “likemylife” shows the largest contribution (0.349 for OLS, 0.305 for GB), followed by other life satisfaction indicators. Although not in the top five, sports-related variables such as “frequencysportsexercise” (physical activity frequency, coded 0–5 from “never” to “daily”) demonstrated notable predictive relevance, supporting the role of physical activity in SWB.

**Figure 3 children-12-01083-f003:**
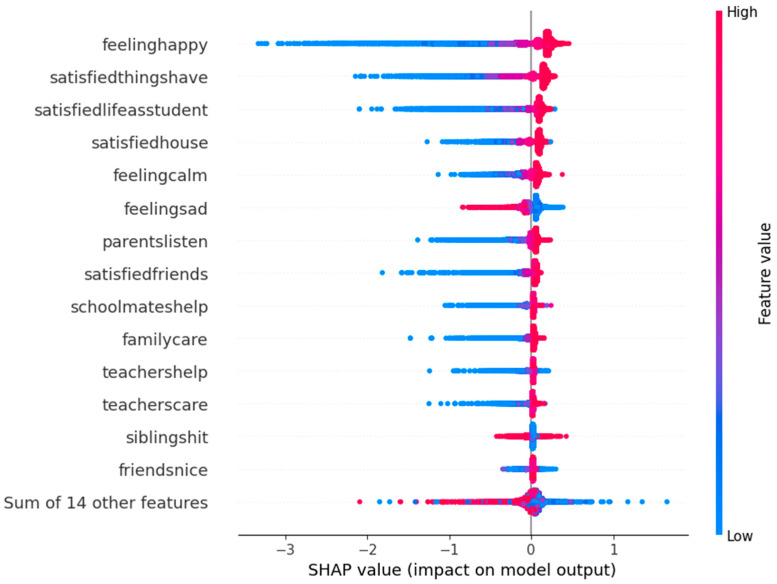
Global importance of variables in predicting subjective well-being (SWB) using the SHAP beeswarm plot. Each point represents an individual observation plotted according to its SHAP value, which indicates the magnitude and direction of that feature’s impact on the model’s prediction (positive values increase predicted SWB; negative values decrease it). Color represents the feature value (red = high, blue = low). For example, high values of “frequencysportsexercise” (coded 0–5 from “never” to “daily”) are predominantly red and aligned with positive SHAP values, indicating a consistent positive association with SWB. This visualization allows interpretation of both the strength and direction of each predictor’s effect across the sample.

**Figure 4 children-12-01083-f004:**
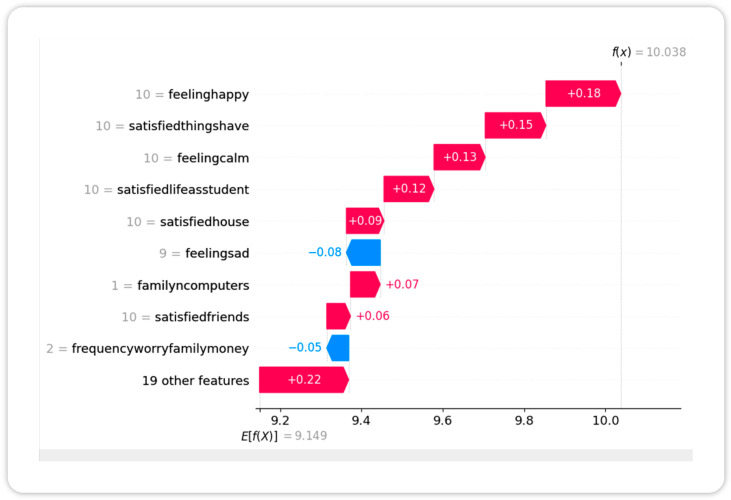
SHAP waterfall plot showing individual prediction decomposition for subjective well-being (SWB). This plot illustrates how specific features contribute to the predicted SWB score for a single child, starting from the model’s base value (average prediction across all samples) and adding or subtracting contributions from individual features. Pink bars indicate variables that increased the prediction, while blue bars indicate those that decreased it. For example, a high score for “frequencysportsexercise” and the presence of “haveequipsportshobbies” pushed the prediction upward, whereas high “feelingsad” had a negative impact. The magnitude of each bar represents the size of that variable’s contribution in points to the final predicted SWB value.

**Figure 5 children-12-01083-f005:**
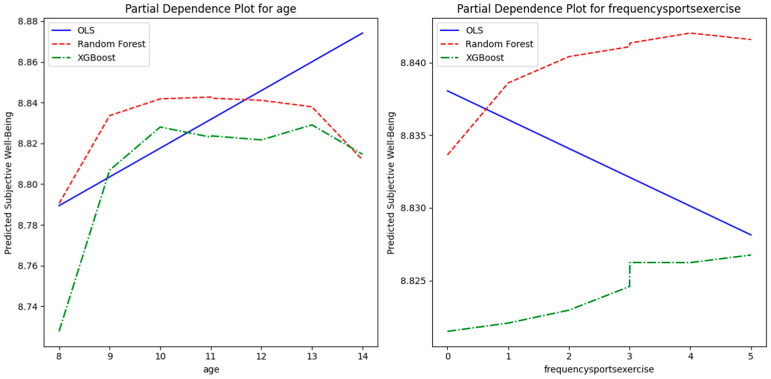
Partial dependence plots of subjective well-being on age and exercise frequency. This graph presents two subplots: the left shows predicted SWB as a function of age (6–14 years), and the right shows predicted SWB as a function of exercise frequency (“frequencysportsexercise”, coded 0–5 for never to daily) across OLS (blue line), Random Forest (red dashed line), and XGBoost (green segmented line). Age exhibits a slight decline, while exercise frequency shows a linear increase with no satiation, illustrating non-linear patterns captured by ML models. Note: Blue line: OLS; red dashed line: Random Forest; green dashed line: XGBoost.

**Table 1 children-12-01083-t001:** Comparison of model performance across restricted and expanded datasets: R^2^ values (95% CI), ΔR^2^ over OLS baseline, MAE, and RMSE. Note: “No leakage” excludes conceptually overlapping affective variables (e.g., “feelinghappy”).

Dataset	Model	R^2^ (95% CI)	ΔR^2^	MAE	RMSE
Restricted	OLS	0.252 (0.23–0.27)	-	0.742	1.323
Restricted	XGBoost (no leakage)	0.35 (0.33–0.37)	+0.098	0.758	1.52
Expanded	XGBoost	0.478 (0.46–0.50)	+0.188	0.745	1.328

## Data Availability

The data that support the findings of this study are publicly available from the Children’s Worlds project website. The dataset from the third wave (ISCWeB 2017–2019) can be accessed at https://isciweb.org/the-data/access-our-dataset/ (accessed on 12 March 2025). The data are publicly available without restrictions for academic purposes after registration.

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
