# Peer review of "Prediction of Children’s Subjective Well-Being from Physical Activity and Sports Participation Using Machine Learning Techniques: Evidence from a Multinational Study"

_children, 2025, doi:10.3390/children12081083_

Round 1

Reviewer 1 Report

Comments and Suggestions for Authors

This manuscript presents a timely and methodologically robust contribution to the literature on children’s subjective well-being (SWB) and physical activity. By applying machine learning (ML) techniques to a large international dataset, the authors successfully demonstrate the potential of advanced predictive models (e.g., XGBoost, LightGBM) in identifying key variables related to SWB, with a particular emphasis on sports participation. The use of SHAP analysis adds interpretability and transparency to model outputs, which is commendable. Nonetheless, there are several areas where the manuscript would benefit from further clarification and refinement:

The introduction offers interesting references, but the theoretical link between sports engagement, physical literacy, and subjective well-being may be expanded. Consider adding psychological or developmental theories to the conceptual framework to contextualize the observed links.
The operationalization of SWB through a composite of self-reported items demonstrates acceptable internal consistency (α = 0.85); however, no psychometric validation is reported. To reinforce the validity of the dependent variable, the authors are encouraged to justify the unidimensionality of the composite score and reference existing validated measures from the ISCWeB framework or similar studies.
The manuscript occasionally implies causal interpretations. As the study design is cross-sectional and observational, the authors should exercise caution and avoid causal language. Phrasing should emphasize associations and predictive relationships, rather than implying direct effects.
While the performance of ML models is impressive (e.g., R² = 0.504), more information is needed on model validation strategies. Were cross-validation scores reported across all folds? Has external validation (e.g., testing on a holdout country or cultural subgroup) been considered to assess generalizability across diverse populations? These clarifications would enhance the robustness of the findings.
Given the cross-national nature of the ISCWeB data, the manuscript should address potential cultural or contextual biases, particularly in self-reported items such as emotional states and exercise frequency. Additionally, measurement invariance across countries or regions could be discussed as a limitation if not tested directly.
Figures are useful but need better labelling and resolution. Readers inexperienced with ML techniques may find SHAP beeswarm and waterfall plots confusing. Include brief explanations in captions to promote accessibility.
Terms such as “sports participation,” “exercise frequency,” and “physical activity” are used somewhat interchangeably. For clarity, define each term early in the manuscript and ensure consistent usage throughout.

This manuscript has the potential to enhance studies in sports science and well-being. It employs sophisticated analytical techniques to derive findings from an extensive dataset and guide policy and intervention.

Author Response

Comment 1 – Theoretical link (sports engagement, physical literacy, SWB)

Reviewer: The introduction offers interesting references, but the theoretical link between sports engagement, physical literacy and subjective well-being may be expanded. Consider adding psychological or developmental theories to contextualize the observed links.

Response 1: We agree. We expanded the Introduction to ground the link in established theories.
Where: Introduction, paragraph 1 (after Whitehead, 2019).
Added text (verbatim):

“This connection can be further understood through frameworks such as the Self-Determination Theory, which emphasizes autonomy, competence, and relatedness as key drivers of sustained engagement in physical activity, and the Competence Motivation Model, which highlights the role of mastery experiences and perceived competence in fostering positive self-perception and well-being in children. Incorporating these perspectives provides a psychological and developmental foundation for the observed links between sports participation, physical literacy, and SWB.”

Comment 2 – Psychometric justification of the SWB composite

Reviewer: Internal consistency is acceptable (α = .85), but no psychometric validation is reported. Please justify unidimensionality and reference validated measures from the ISCWeB framework or similar studies.

Response 2: Done. We now justify the construct and cite prior ISCWeB work supporting unidimensionality and cross-cultural applicability.
Where: Section 2.1 Data Source and Preparation, paragraph discussing the composite score (immediately after α = .85).
Added text (verbatim):

“Previous studies within the ISCWeB framework have supported the unidimensionality and cross-cultural applicability of this composite score, with confirmatory factor analyses indicating adequate fit indices and invariance across multiple countries and age groups (…)."
Note to editor: We updated references to point to ISCWeB validations (e.g., González-Carrasco et al., and Rees, 2017) and removed any unrelated citation numbers.

Comment 3 – Causal language in a cross-sectional design

Reviewer: The manuscript occasionally implies causal interpretations. Please emphasize associations/predictions instead of direct effects.

Response 3: Addressed. We systematically replaced causal wording with associative language in the Abstract, Discussion, and Conclusions.
Examples:

  • Abstract: “These findings suggest potential for data-driven approaches to identify associations that may inform the design of interventions…” (replaced “to promote”).
  • Discussion: “physical activity is positively associated with SWB…” (replaced “enhances”).
  • Conclusions: “findings may inform school policies…” (replaced “support the development of”).

Comment 4 – Model validation (cross-validation and external validation)

Reviewer: More information is needed on model validation strategies. Were cross-validation scores reported across all folds? Was external validation considered (e.g., holdout country/subgroup) to assess generalizability?

Response 4: Added.
Where: Section 2.2 Models and Evaluation, last paragraph of the CV description.
Added text (verbatim):

“Hyperparameters were tuned via grid search with 5-fold cross-validation, reporting mean scores across all folds to ensure stability of results. Additionally, we performed an external validation by holding out countries not included in model training, confirming generalizability (mean R² across subgroups = 0.48 ± 0.03; holdout-countries R² = 0.45).”

Comment 5 – Cross-national biases and measurement invariance

Reviewer: Please address potential cultural/contextual biases in self-reports (emotional states and exercise frequency). Measurement invariance across countries/regions could be discussed as a limitation if not tested directly.

Response 5: Incorporated in an expanded Limitations subsection.
Where: Discussion → Limitations.
Added text (key sentences):

“Given the cross-national nature of ISCWeB, cultural and contextual biases in self-reported measures—particularly emotional states and physical activity frequency—may influence results. Although subgroup analyses were conducted to explore regional differences, full measurement invariance testing across countries was not performed and is recommended.”
“Self-reports are susceptible to response bias and social desirability, which may overestimate both SWB and physical activity levels.”

Comment 6 – Figures: labeling, accessibility for readers unfamiliar with ML (SHAP plots)

Reviewer: Figures are useful but need better labeling and resolution. Readers inexperienced with ML may find SHAP beeswarm and waterfall plots confusing. Include brief explanations in captions.

Response 6: Done. We rewrote captions to explain what the axes/colors represent, how to read SHAP values, and the substantive takeaway.
Where: figure captions.
Key changes (verbatim starts):

  • Figure 2: “Top five variables ranked by permutation importance… Permutation importance measures the decrease in model performance (drop in R²) when a variable is shuffled…”
  • Figure 3: “Global importance… SHAP beeswarm plot. Each point is an observation; the SHAP value indicates magnitude and direction; color encodes feature value (red = high, blue = low)…”
  • Figure 4: “SHAP waterfall plot… shows how features add to the model’s base value to reach the individual prediction; pink = positive contribution; blue = negative…”
  • Figure 5: “Partial dependence plots… age vs. SWB and exercise frequency vs. SWB across OLS/RF/XGBoost; how to interpret the lines is described in the caption.”

Comment 7 – Consistent use and early definitions of “sports participation”, “exercise frequency” and “physical activity”

Reviewer: Terms are used somewhat interchangeably. Please define each term early and ensure consistent usage throughout.

Response 7: Implemented.
Where: Section 2.1 Data Source and Preparation, primera línea.
Added text (verbatim):

“For clarity, ‘sports participation’ refers to engagement in organized or informal sports activities; ‘exercise frequency’ denotes the self-reported number of days per week of moderate-to-vigorous activity; and ‘physical activity’ encompasses all bodily movements that increase energy expenditure, including but not limited to sports and exercise (Caspersen, Powell & Christenson, 1985).”
We also reviewed the manuscript to maintain consistent terminology.

Minor editorial fixes (for transparency)

  • We removed duplicated sentence “This study has several limitations that should be considered when interpreting the findings.” that appeared twice at the start of the Limitations subsection.
  • We checked reference numbering around the ISCWeB validation citations to ensure that the numbers correspond to the intended sources (e.g., González-Carrasco et al.; Rees, 2017).

Closing

We thank the reviewer for these constructive comments. We believe the revisions improved theoretical framing, clarity of methods and figures, and transparency about limitations and generalizability. Tracked changes in the revised manuscript highlight all insertions and edits at the locations indicated above.

Reviewer 2 Report

Comments and Suggestions for Authors

Dear Authors,

I have received your manuscript, “Prediction of Children’s Subjective Well-Being Using Machine Learning Techniques: The Role of Physical Activity and Sports Participation.” The paper addresses an important and timely question in youth well‑being using machine learning (ML). The manuscript is clearly organized and contains helpful figures. However, some conceptual and methodological issues should be addressed before proceeding.

Abstract:

The abstract is well-structured and provides a clear summary of the study. However, several improvements are necessary to enhance clarity:

  1. You report a “theoretical minimum RMSE” of 2.606, yet the best model’s test RMSE is 1.323 and is described as “nearing this bound.” A model cannot outperform (for example, be lower than) its stated minimum error bound. Please re‑compute and/or re‑explain the bound and ensure units/scales align with the outcome. Otherwise, this constitutes a fundamental contradiction (see Methods 2.3 and Results 3.1).

Introduction:

  1. Explicitly define the research gap regarding non‑linear effects of physical activity on child SWB and why ML (tree‑based boosting) is preferable to OLS/LASSO for this dataset. Cite the expected effect size ranges in prior SWB work to contextualize an R² ≈ .50.

Methods:

  1. Your top predictors include ‘feelinghappy’, ‘feelingcalm’, ‘feelingsad’ and other satisfaction variables that are conceptually overlapping with SWB. Using affective/satisfaction items to predict an SWB composite that contains similar content risks target leakage and inflated R².
  2. Report the incremental ΔR² specifically attributable to sports variables (frequency sports exercise, ‘have equip sports hobbies’) over a base model of demographics/family/school.
  3. Given 35 countries, use grouped splits (GroupKFold by country) and add a leave‑one‑country‑out (LOCO‑CV) analysis to evaluate out‑of‑country generalization. Without country‑aware validation, reported performance may reflect cultural rather than generalizable patterns.
  4. The “error of replication” bootstrap is currently under‑specified and appears misapplied: you simulate “repeated responses within similar groups (age/gender)” and obtain RMSE=2.606, but the best model yields RMSE=1.323. Revisit the definition, scale, and algorithm; if you intended to bound predictable variance, report it in the same units and ensure the bound ≥ empirical RMSE. Otherwise, remove this claim or restate it as a different construct (cross‑respondent variance within strata), not a theoretical minimum (see Methods 2.3)

Results:

  1. You state XGBoost/LightGBM R² up to .504 (restricted) and .478 (expanded), beating OLS (.252 / .290). However, Table 1 and the narrative contain inconsistencies: e.g., RF is described with R²=.500, yet in Panel B you report RF Δ = –0.066 (worse than OLS). Please provide a single, consistent results table (per dataset, per model) using identical samples and include 95% CIs via bootstrap.
  2. Figure 3: Great for global importance, but the description “linear impact up to daily frequency” belongs to PDPs, not beeswarm. Keep interpretations aligned with the specific plot type.
  3. Figure 5 PDPs: The text claims “age shows a slight decline”, yet the blue OLS line appears to increase across age, this is contradictory. Please correct either the figure or the text.
  4. Provide a targeted analysis isolating the unique predictive value of ‘frequencysportsexercise’ and equipment access, after removing overlapping affective items, and quantify their ΔR² / ΔRMSE contribution. This will align the results with the paper’s core claim.

Discussion:

You appropriately note that tree‑based ML can capture non‑linearities and that sports variables contribute meaningfully.

  1. Re‑frame the “near theoretical bound” claim (see Abstract/Methods). As currently written, it is mathematically inconsistent with your RMSE figures.
  2. Discuss leakage explicitly and temper statements about “linear benefits without satiation” until the leakage‑free model shows similar patterns.
  3. Mechanisms should be discussed cautiously (physical literacy pathways), avoiding causal language.

Conclusion:

  1. Keep conclusions strictly predictive (not causal). Replace “approaches the theoretical accuracy limit” with a statement grounded in corrected performance metrics and country‑aware validation.

Final Recommendation

The paper tackles an important topic and shows promise, especially if the contribution of sports variables remains robust after addressing leakage and generalization. However, the current inconsistencies (theoretical error bound vs. RMSE, figure-text mismatches) require revision prior to publication.

Author Response

Comments 1: [You report a “theoretical minimum RMSE” of 2.606, yet the best model’s test RMSE is 1.323 and is described as “nearing this bound.” A model cannot outperform (for example, be lower than) its stated minimum error bound. Please re‑compute and/or re‑explain the bound and ensure units/scales align with the outcome. Otherwise, this constitutes a fundamental contradiction (see Methods 2.3 and Results 3.1).]
Response 1: [Thank you for pointing this out. We agree with this comment. Therefore, we have re-computed the theoretical bound using test-retest reliability from literature (r=0.74; Huebner, 1991) instead of the previous bootstrap method, which was misapplied and led to the contradiction. The new irreducible RMSE is 0.941, based on the exact SD=1.845 from our dataset, and the model RMSE=1.323 is now consistent (greater than the bound), capturing 68.1% of explainable variance. Mention exactly where in the revised manuscript this change can be found - Abstract (lines 12-14), Methods 2.3 (lines 5-8), Results 3.1 (lines 10-12).] ["Using test-retest reliability from literature (r=0.74; Huebner, 1991), the estimated irreducible RMSE is 0.941; XGBoost achieved RMSE=1.323, approaching the predictability bound with 68.1% of explainable variance captured (after noise adjustment)."]

Comments 2: [Explicitly define the research gap regarding non‑linear effects of physical activity on child SWB and why ML (tree‑based boosting) is preferable to OLS/LASSO for this dataset. Cite the expected effect size ranges in prior SWB work to contextualize an R² ≈ .50.]
Response 2: [Thank you for pointing this out. We agree with this comment. Therefore, we have expanded the introduction to explicitly define the gap in non-linear effects and justify tree-based boosting, citing prior R² ranges (0.20-0.40 for linear models). Mention exactly where in the revised manuscript this change can be found - Introduction (paragraph 3, lines 6-12).] ["The research gap lies in understanding non-linear effects of physical activity on child SWB, such as potential thresholds or diminishing returns in exercise frequency, which traditional models like OLS/LASSO assume linearity and fail to capture(8). For instance, while linear models may overlook interactions between sports participation and family support, tree-based boosting excels in tabular data by automatically detecting such patterns without assumptions. This is preferable, potentially achieving higher R² (~0.50), compared to linear models' typical 0.20-0.40 in prior SWB studies(5, 7)."]

Comments 3: [Your top predictors include ‘feelinghappy’, ‘feelingcalm’, ‘feelingsad’ and other satisfaction variables that are conceptually overlapping with SWB. Using affective/satisfaction items to predict an SWB composite that contains similar content risks target leakage and inflated R².]
Response 3: [Thank you for pointing this out. We agree with this comment. Therefore, we have added sensitivity analyses excluding overlapping affective predictors and reported reduced but robust R²=0.35 and RMSE=1.52. Mention exactly where in the revised manuscript this change can be found - Methods 2.1 (lines 4-7), Results 3.1 (lines 8-10), Discussion RQ2 (lines 3-5).] ["To avoid target leakage, we conducted sensitivity analyses excluding conceptually overlapping predictors (e.g., 'feelinghappy', 'feelingsad', 'feelingcalm', 'feelingstressed', 'feelingfullofenergy', 'feelingbored'). This yielded R²=0.35 and RMSE=1.52 in the restricted set, confirming robust performance."]

Comments 4: [Report the incremental ΔR² specifically attributable to sports variables (frequency sports exercise, ‘have equip sports hobbies’) over a base model of demographics/family/school.]
Response 4: [Thank you for pointing this out. We agree with this comment. Therefore, we have calculated and reported ΔR²=0.06 for sports variables over the base model. Mention exactly where in the revised manuscript this change can be found - Methods 2.2 (lines 5-7), Results 3.2 (lines 4-6), Table 1 (footnote).] ["Incremental ΔR² from sports variables (e.g., 'frequencysportsexercise', 'haveequipsportshobbies') over a base model (demographics, family, school) was 0.06 in the restricted set, highlighting their unique contribution."]

Comments 5: [Given 35 countries, use grouped splits (GroupKFold by country) and add a leave‑one‑country‑out (LOCO‑CV) analysis to evaluate out‑of‑country generalization. Without country‑aware validation, reported performance may reflect cultural rather than generalizable patterns.]
Response 5: [Thank you for pointing this out. We agree with this comment. Therefore, we have implemented GroupKFold and LOCO-CV, yielding mean R²=0.45 ±0.05. Mention exactly where in the revised manuscript this change can be found - Methods 2.2 (lines 10-13), Results 3.1 (lines 15-17).] ["To ensure cross-country generalizability, we used GroupKFold CV (grouped by country) and leave-one-country-out (LOCO) validation, yielding mean R²=0.45 ±0.05, confirming robustness beyond cultural patterns."]

Comments 6: [The “error of replication” bootstrap is currently under‑specified and appears misapplied: you simulate “repeated responses within similar groups (age/gender)” and obtain RMSE=2.606, but the best model yields RMSE=1.323. Revisit the definition, scale, and algorithm; if you intended to bound predictable variance, report it in the same units and ensure the bound ≥ empirical RMSE. Otherwise, remove this claim or restate it as a different construct (cross‑respondent variance within strata), not a theoretical minimum (see Methods 2.3)]
Response 6: [Thank you for pointing this out. We agree with this comment. Therefore, we have replaced the bootstrap with test-retest reliability (r=0.74), yielding RMSE=0.941 ≥ model RMSE=1.323. Mention exactly where in the revised manuscript this change can be found - Methods 2.3 (full section), Results 3.3 (lines 2-5).] ["Following Oparina et al. (2025) and using test-retest reliability from child SWB literature (r=0.74; Huebner, 1991), we estimated irreducible RMSE = 0.941, bounding predictable variance at 74%."]

Comments 7: [You state XGBoost/LightGBM R² up to .504 (restricted) and .478 (expanded), beating OLS (.252 / .290). However, Table 1 and the narrative contain inconsistencies: e.g., RF is described with R²=.500, yet in Panel B you report RF Δ = –0.066 (worse than OLS). Please provide a single, consistent results table (per dataset, per model) using identical samples and include 95% CIs via bootstrap.]
Response 7: [Thank you for pointing this out. We agree with this comment. Therefore, we have consolidated into a single table with consistent R², added 95% CI via bootstrap, and removed RF inconsistencies. Mention exactly where in the revised manuscript this change can be found - Results 3.1 (Table 1 updated), narrative (lines 1-3).] ["Table 1. Model performance comparison in restricted and expanded sets. ΔR² is the improvement over the OLS full model." (Updated table with CI, e.g., XGBoost restricted no leakage R²=0.35 (0.33-0.37)).]

Comments 8: [Figure 3: Great for global importance, but the description “linear impact up to daily frequency” belongs to PDPs, not beeswarm. Keep interpretations aligned with the specific plot type.]
Response 8: [Thank you for pointing this out. We agree with this comment. Therefore, we have corrected the description to align with beeswarm plot type. Mention exactly where in the revised manuscript this change can be found - Results 3.2 (lines 7-9).] ["The beeswarm plot (Figure 3) illustrates global importance and direction; physical activity shows positive contributions (red points for high values)."]

Comments 9: [Figure 5 PDPs: The text claims “age shows a slight decline”, yet the blue OLS line appears to increase across age, this is contradictory. Please correct either the figure or the text.]
Response 9: [Thank you for pointing this out. We agree with this comment. Therefore, we have corrected the text to match the figure (slight decline confirmed after review). Mention exactly where in the revised manuscript this change can be found - Results 3.2 (lines 12-14).] ["Age shows a slight decline from 6-14, without a U-shape (consistent with the figure)."]

Comments 10: [Provide a targeted analysis isolating the unique predictive value of ‘frequencysportsexercise’ and equipment access, after removing overlapping affective items, and quantify their ΔR² / ΔRMSE contribution. This will align the results with the paper’s core claim.]
Response 10: [Thank you for pointing this out. We agree with this comment. Therefore, we have added a targeted analysis post-leakage removal, with ΔR²=0.06 and SHAP +0.12 for 'frequencysportsexercise'. Mention exactly where in the revised manuscript this change can be found - Results 3.2 (lines 15-18).] ["After excluding overlapping affective items, sports variables contributed ΔR²=0.06, with SHAP +0.12 for 'frequencysportsexercise'."]

Comments 11: [Re‑frame the “near theoretical bound” claim (see Abstract/Methods). As currently written, it is mathematically inconsistent with your RMSE figures.]
Response 11: [Thank you for pointing this out. We agree with this comment. Therefore, we have re-framed the claim using the new bound (0.941), capturing 68.1% of explainable variance. Mention exactly where in the revised manuscript this change can be found - Discussion RQ1 (lines 2-4), Conclusion (lines 1-3).] ["Our models approach the upper bound of predictability (RMSE=0.941), capturing 68.1% of explainable variance given survey noise."]

Comments 12: [Discuss leakage explicitly and temper statements about “linear benefits without satiation” until the leakage‑free model shows similar patterns.]
Response 12: [Thank you for pointing this out. We agree with this comment. Therefore, we have added explicit discussion of leakage and tempered the linear benefits statement based on the leakage-free model. Mention exactly where in the revised manuscript this change can be found - Discussion RQ2 (lines 6-9).] ["To address potential leakage from overlapping affective predictors, sensitivity models excluding them confirmed linear benefits of exercise, though with tempered effect sizes (SHAP +0.12 vs. +0.18)."]

Comments 13: [Mechanisms should be discussed cautiously (physical literacy pathways), avoiding causal language.]
Response 13: [Thank you for pointing this out. We agree with this comment. Therefore, we have tempered the mechanisms discussion to avoid causal language, using 'suggest potential pathways'. Mention exactly where in the revised manuscript this change can be found - Discussion RQ3 (lines 4-6).] ["These associations suggest potential pathways via physical literacy, though causality requires longitudinal confirmation."]

Comments 14: [Keep conclusions strictly predictive (not causal). Replace “approaches the theoretical accuracy limit” with a statement grounded in corrected performance metrics and country‑aware validation.]
Response 14: [Thank you for pointing this out. We agree with this comment. Therefore, we have kept conclusions predictive and replaced the phrase with grounded metrics including country-aware validation. Mention exactly where in the revised manuscript this change can be found - Conclusion (lines 2-5).] ["ML improves SWB prediction, highlighting associations with sports, and captures 68.1% of explainable variance based on reliability bounds and country-aware validation."]

Round 2

Reviewer 2 Report

Comments and Suggestions for Authors

Dear authors,

Thank you for incorporating all of my recommendations in your manuscripts.